# Repeated exposure to nanosecond high power pulsed microwaves increases cancer incidence in rat

**René de Seze**[1]*, **Carole Poutriquet**[2], **Christelle Gamez**[1], **Emmanuelle Maillot-Maréchal**[1], **Franck Robidel**[1], **Anthony Lecomte**[1], **Caroline Fonta**[2]

**1** Chronic Risks Division, PeriTox/Experimental Toxicology Unit UMR-I 01, Institut National de l'Environnement Industriel et des Risques, Verneuil en Halatte, France, **2** Brain and Cognition Research Center CerCo, Centre National de la Recherche Scientifique UMR5549, Université de Toulouse, Toulouse, France

* rene.de-seze@ineris.fr

**Data Availability Statement:** All relevant data are within the paper and its Supporting Information files.

## Abstract

High-power microwaves are used to inhibit electronics of threatening military or civilian vehicles. This work aims to assess health hazards of high-power microwaves and helps to define hazard threshold levels of modulated radiofrequency exposures such as those emitted by the first generations of mobile phones. Rats were exposed to the highest possible field levels, under single acute or repetitive exposures for eight weeks. Intense microwave electric fields at 1 MV m$^{-1}$ of nanoseconds duration were applied from two sources at different carrier frequencies of 10 and 3.7 GHz. The repetition rate was 100 pps, and the duration of train pulses lasted from 10 s to twice 8 min. The effects on the central nervous system were evaluated, by labelling brain inflammation marker GFAP and by performing different behavioural tests: rotarod, T-maze, beam-walking, open-field, and avoidance test. Long-time survival was measured in animals repeatedly exposed, and anatomopathological analysis was performed on animals sacrificed at two years of life or earlier in case of precocious death. Control groups were sham exposed. Few effects were observed on behaviour. With acute exposure, an avoidance reflex was shown at very high thermal level (22 W kg$^{-1}$); GFAP was increased some days after exposure. Most importantly, with repeated exposures, survival time was 4-months shorter in the exposed group, with eleven animals exhibiting a large sub-cutaneous tumour, compared to two in the sham group. A residual X-ray exposure was also present in the beam (0.8 Gy), which is probably not a bias for the observed result. High power microwaves below thermal level in average, can increase cancer prevalence and decrease survival time in rats, without clear effects on behaviour. The parameters of this effect need to be further explored, and a more precise dosimetry to be performed.

**Funding:** This study was funded by the French General Directorate of Army and the research program DRC07-AP05 of the Ministry of Ecology. The General Directorate of Army agreed the study design.

**Competing interests:** The authors have declared that no competing interests exist.

## Introduction

High power microwaves (HPM) are used to inhibit the electronic systems of threatening vehicles. Concern has arisen as to whether HPM could lead to health hazards for operators of emitting systems and for staff members exposed in targeted vehicles. The health effects of HPM have been studied since the discovery of radar in the middle of the past century. Many experiments have been performed with microsecond pulses at levels of several hundred kilovolts per meter. Some studies have been published, but many others have been presented only as reports or at scientific meetings.

Studies performed with a whole-body specific absorption rate (SAR) above the thermal threshold of 4 W kg$^{-1}$ have shown biological effects. Below 4 W kg$^{-1}$, for studies showing effects, the pulse duration of single pulses was between 40 ns and 10 μs, and temporal peak-SAR was from 5 to 20 MW kg$^{-1}$. Half of the studies on HPM addressed behavioural end-points, reviewed by D'Andrea [1]. Others bear on the cardio-vascular [2,3], visual [4], and auditory systems [5]. Only sparse work concerned blood-brain-barrier permeability [6,7], DNA damage [8,9], carcinogenesis [10,11], or cellular or sub-cellular mechanisms [12]. Another paper by Dorsey et al. [13] on carcinogenesis used ultrawideband (UWB) exposure. Concerning cancer, Zhang et al. [14] and Devyatkov et al. [10] reported protective effects at levels above thermal threshold, with smaller tumours and a 30% increase of survival rate in the exposed group. Focusing on UWB exposures, several years after Seaman' article [15], a recent review by Schunck et al. reported only one new paper in 2009, and no other effects on cancer were reported [16]. However, durations of exposure in those studies were often short.

Using a realistic source of HPM, this study investigated whether the highest possible exposure levels could produce behavioural or functional effects in rats acutely exposed, or chronic pathological effects with a repetitive exposure for two months. We assessed effects of 3.7 and 10 GHz nanosecond pulsed HPM around 1 MV m$^{-1}$ on the health and lifespan of male Sprague-Dawley rats.

## Methods

### Literature survey

We looked at the scientific and medical literature (NCBI-PubMed, Current Contents and Science Direct, more recently Web of Science) and at specialized databases of papers, scientific meetings and reports (EMF Database, IEEE ICES EMF Literature Database and WHO- EMF--Portal). The following keywords were used: HPM, high power microwave, high peak, electromagnetic pulse, microwave radiation, high exposure microwave, HPPP, EHPP, high intensity microwave.

### Exposure systems

The sources of high-power microwaves (HPM) were two superradiance generators, one in X-band at 10 GHz (SRX) with pulses of 1 ns, the other in S-band at 3.7 GHz (SRS) with pulses of 2.5 ns. The strongest possible microwave electric fields were applied, of about 1 MV m$^{-1}$, at a repetition rate of 100 pps (Table 1). The "SINUS type" electron accelerator of this system is made of a Tesla generator and a continuous formation line. A great advantage of this system is its small size. The superradiance source is derived from the back-wave oscillator, with the following characteristic features: ultrashort microwave pulses, and very high peak power. Pulses were gated sinewaves, with a bandwidth around 10% and rise/fall-times of about 0.2ns.

**Table 1. Exposure parameters of the two HPM sources.**

|  | SRX | SRS acute | SRS avoidance | SRS repeated |
|---|---|---|---|---|
| **Beam diameter at output** | 13 cm | 22 cm | | |
| **Frequency** | 10 GHz | 3.7 GHz | | |
| **Total emitting power** | 350 MW | 500 MW | | |
| **Pulse duration** | 1 ns | 2.5 ns | | |
| **Train duration** | 10 s | Continuous | | |
| **Emission duration** | Every 5 min for 1 h | 2 x 8 min | 14 min | 2 x 8 min |
| **Peak surface power at output** | 20 GW m$^{-2}$ | 2 GW m$^{-2}$ | | |
| **Distance from the horn** |  | 0.60 m | 0.13 m | 2.5 m |
| **Peak E-field** | 3 MV m$^{-1}$ | 1.7 MV m$^{-1}$ | 2.9 MV m$^{-1}$ | 0.56 MV m$^{-1}$ |
| **Peak SAR** | 95 MW kg$^{-1}$ | 31 MW kg$^{-1}$ | 90 MW kg$^{-1}$ | 3.33 MW kg$^{-1}$ |
| **Average SAR over total exposure** | 0.34 W kg$^{-1}$ | 4.7 W kg$^{-1}$ | 22 W kg$^{-1}$ | 0.83 W kg$^{-1}$ |

## Animals

Six-weeks-old Sprague Dawley male rats were purchased from Charles River, L'Arbresle, France. Identification of animals was made with a marker pen on the tail for acute studies, or by tattoo on the ear pavilion during the long-term follow-up. Rats were accustomed to the animal facility for 5 days before handling. They were kept under specific pathogen free (SPF) controlled environmental conditions (ambient temperature 22±1 ◦C, 12-h light/12-h dark cycle) and received food and tap water ad libitum, except during exposure. Rats were housed in groups of 2 animals per enriched cage for social comfort. Enrichment consisted in adding corn chips for animals to nest and play. The day before exposure, rats were handled and accustomed to the experimenter. Further, they were either exposed (N = 150) or sham-exposed (N = 144). The protocol was reviewed and approved by INERIS ethical committee. Animals were monitored clinically and for mortality once a day. Closer surveillance was performed in case of clinical observations, such as behaviour changes, dull hair or upon appearance of a tumour. In the study with repetitive exposure, the criteria to determine when animals should be ethically euthanized during the follow-up were: weight loss (more than 20% as compared to the week before), ulceration of a tumour, tumour size larger than 8 cm, impaired movement, loss of spontaneous activity or loss of reactions to stimulus.

Animal dedicated to immunohistochemical analyses, to avoid pain and distress, were injected with an overdose of pentobarbital (i.p.; 50 mg kg$^{-1}$) until brain-dead stage.

## Exposure protocol

Two types of acute exposures were carried out. The SRX exposure lasted 10 s every 5 min for one hour, and the SRS exposure lasted 2 x 8 min with 10 min interval (26 min total). Rats were transferred in plastic boxes with filtering covers from the animal facility through an airlock to the shielded exposure room 50 m apart. For acute exposures, rats were exposed one by one in their cage directly at the horn output (246 animals in total, 12 per group, except one group of 6 for preliminary viability tests). Besides, one protocol of repeated exposures was used with SRS source for 48 rats. The 26 minutes-exposure was repeated each day, 5 days/week for 8 weeks. When performing mean term repetitive exposures every day with a realistic source that cannot easily be duplicated, there is a need for optimization of the design to expose many animals at the same time. The circular beam produced by the TM01 mode of the waves was adapted to this goal, with a beam width large enough to simultaneously expose 12 animals at 2.5 m from the SRS output. Animals were exposed 2 by cage in six cages placed each day at different

positions on the circle. Then every day, 2 series of 12 animals were exposed, alternating with 2 series of sham exposure in-between to allow time for the equipment to cool down between two successive exposure sessions. In total, two groups of 24 rats received a repeated exposure, either real, either sham.

After the end of repeated exposures, the animals were observed and followed up to 2 years of age. Lifespan was recorded, and anatomo-pathological examination was performed at the animal death.

For each test, a group of 12 exposed animals was compared to a similar-sized group of sham-exposed animals, put in the same place and under the same ambient conditions than the exposed animals, but without emission from the source.

## Investigations on the central nervous system

After one acute or the last repetitive exposure, different behavioural tests were performed blindly: beam-walking, rotarod, T-maze, open-field. An avoidance test was also performed during an acute SRS exposure, applied continuously for 14 minutes. In the avoidance test, animals can choose to spend time in two parts of a box. One part is protected against the beam (shielded), the other is not. Animals behaviour is recorded with a video camera. Upon analysis of the movies, the time spent in the non-protected side is recorded.

After the behavioural tests were performed, animals were sacrificed 2 days after the SRX and the SRS exposures, and 7 days after the SRX exposure, and an immunohisto-chemical labelling of the brain inflammation marker GFAP was achieved on 40 μm thick slices for 5 parts or regions of the brain: frontal cortex, gyrus dentate, putamen, pallidum and cerebellar cortex.

The global design of this study and the sample size in each test are summarized in Table 2.

## Immunohistology

**Brain preparation.** An intra-cardiac perfusion was performed with a 0.9% NaCl solution, followed by a 4% paraformaldehyde solution in 0.1 M phosphate buffer (pH 7.6) at a flow rate of 30 ml min$^{-1}$. The brain was dissected and immerged in a 30% sucrose/4% paraformaldehyde solution for 48 h under agitation at 4 ˚C to cryo-protect the cerebral tissue Using a freezing

**Table 2. Global design of the study.**

|  | SRX | SRS | |
|---|---|---|---|
| Exposure | Acute | acute | repeated |
| Emission duration | 10 s every 5 min for 1 h | 2 x 8 min with 10 min interval Avoidance: 14 min continuous | 2 x 8 min with 10 min interval 5 days/ week for 8 weeks |
| Age at experiment | 6 weeks | 6 weeks | 6 weeks |
| N animals/exposition | 1 | 1 | 12 |
| Behavioural tests | Beam walking (n = 13/11[a]), rotarod (n = 12/12), T-maze (13/11), open field (n = 12/12), avoidance (n = 11/10) | Beam walking, rotarod, T-maze, avoidance (n = 12/12) | T-maze (n = 8/8, wk14[b]), Beam walking (n = 12/12, wk15), rotarod (n = 24/24, wk16) |
| GFAP staining (J = exposition day n = nb animals) | J+2n = 12/12 J+7 n = 11/11 | J+2 n = 12/12 | |
| Anatomo-pathology (HES) | 104 weeks or at death | 104 weeks or at death | 104 weeks or at death |
| Lifespan recording (up to 2 years) | | | n = 24/24 |

[a]n = n1/n2 = [number of exposed animals] / [number of sham animals].

[b]wk = age of animals at the date of test.

microtome, 40 μm-thick sagittal brain slices were obtained and processed as free-floating sections for GFAP immunodetection [17].

**Immunohistochemistry.**   Brain slices were washed in Phosphate Buffer Saline (PBS; pH 7.6) 3×20 min and incubated for 30 min at room temperature in 1% H2O2 to block endogenous peroxidases. Background noise was saturated by incubation for one hour in a solution of goat serum (5%), bovine serum albumin (2%) and Triton (0.3% Triton). Slices were then incubated overnight at 4 ∘C with the primary polyclonal antibody anti-GFAP (rabbit antibody, 1/8000 in PBS). On the second day, the slices were washed with PBS-Triton (3×20 min) and incubated for 1 h with the secondary antibody (biotinylated goat anti-rabbit, ABC kit 1/500 in PBS). Sections were washed in PBS (3x20 min). Immunostaining was revealed with an ABC/VIP kit (Vectastain, Vector). Slices were mounted on slides, dehydrated then mounted with DPX before image analysis. Surfaces of labelled areas have been quantified using an optical microscope (Zeiss) coupled with a Colour Camera 3 Charge Coupled Device (CCD) (Sony) and the Visilog 6.2 (NOESIS society, Les Ulis, France) analyser system.

## Anatomopathology

Animals involved in repeated exposures were followed up to 2 years of age. Individual lifespan was recorded for animals needing an ethical euthanasia. Either at this time or at 104 weeks of age, animals were sacrificed with a lethal overdose of pentobarbital (5.0 ml kg$^{-1}$ IP), organs were collected and cut into smaller pieces if larger than 5 mm in order to optimize tissue fixation in 4% buffered formalin for 48 to 72h. All tissue samples were included in paraffin blocks.

Five μm slices were obtained with a microtome and two slides per organ were prepared. An anatomo-pathological examination was performed on one slide per organ, that was coloured with haematoxylin-eosin stain (HES). The second section was stored in case of need for any other specific labelling.

## Dosimetry

Electric field has been measured at the actual exposure distance of 2.5 m from the source output with a germanium detector and calculated for closer distances. As numerical computation of specific absorption rate (SAR) by FDTD was not available, the whole-body specific absorption rates (SAR) (defined as electromagnetic power absorbed per unit of tissue mass) were calculated for each condition from the rat's position and size as described by Gandhi [18] and Durney et al. [19]. Time-averaged SARs were 0.83 W kg$^{-1}$ for the repeated exposure, and between 0.34 and 22 W kg$^{-1}$ for acute exposures. Temporal peak SARs during the pulses ranged between 3.3 and 95 MW kg$^{-1}$ (Table 1). Some residual X-rays came out from the device, for 20 mGy/day (total 0.8 Gy). Numerical and experimental dosimetry and thermometry are needed to strengthen the results of this study.

## Statistics

Percentages of time spent in the exposed or in the blinded box (avoidance study) were compared by two-way ANOVA with two factors: exposure (exposed or sham) and period (habituation or exposure). Percentages of labelled areas for GFAP were compared by two-way ANOVA with two factors: exposure (exposed or sham) and localisation (brain area) with R software (Version 2.5.0). Survival rates of repetitively exposed animals were compared with Prism 5 v5.02 by the log-rank test (Mantel-Cox), with calculation of two-tail p value, of the median survival and of the hazard ratio between both groups by the Mantel-Haenszel method.

## Results

### Behavioural tests

First, behavioural tests to evaluate cognitive and sensori-motor functions were performed. No effects were observed after acute or repetitive exposures on behavioural results in beam-walking, T-maze and open-field tests. After repeated exposure to a superradiance S source (SRS) of HPM, rotarod performance was assessed: rats had to stay for three minutes on an axis rotating at 16 turns per minute. They underwent one training session and one test session. In the test session, rotarod performance was significantly enhanced in exposed animals: 41/72 exposed animals succeeded, compared to 23/72 sham animals—p < 0.001 (S1 Table).

During an acute SRS exposure, with a choice for rats to stay in an exposed or a shielded compartment (avoidance test), exposed animals spent 3.7% ± 3.5% of time on the exposed side, compared to 21.9% ± 9.1% for the sham group (mean ± SD; p < 0.001) (S1 Fig and S2 Table).

### Brain inflammation

Then, brain inflammation was assessed by measuring glial fibrillary acidic protein (GFAP) levels, indicative of damaged or dysfunctional cerebral tissue. With a superradiance X source (SRX) of HPM, expression of GFAP was not increased two days (D2), but was increased seven days (D7), after an acute exposure, in all brain areas, except the cerebellum cortex (+50.0% ± 7%—mean ± SD; p < 0.02). With SRS, GFAP expression was increased two days after acute exposure in all brain areas (D2 - +112% ± 19%—mean ± SD; p < 0.001) (Fig 1 and S3 and S4 Tables).

### Lifespan

Most strikingly, six exposed animals deceased early between 33 and 47 weeks, leading to a 4-months decrease of lifespan in the repetitively exposed group (n = 24) compared to the sham

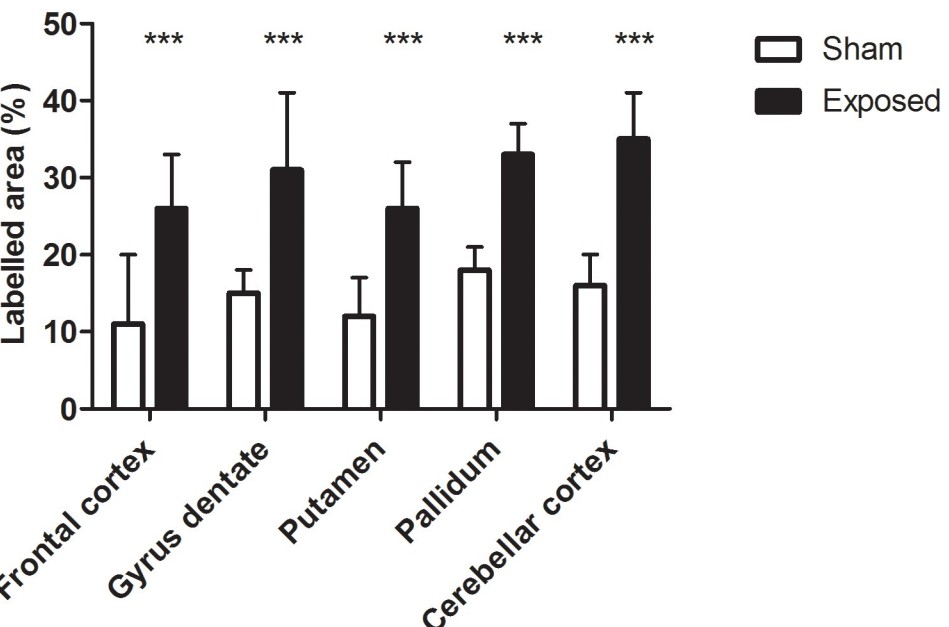

**Fig 1. GFAP expression after repeated exposure to SRS source.** GFAP immunohistochemical labelling in different brain areas two days after exposure with Superradiance S source (% labelled area—Mean ± SD). White = sham (n = 12); black = exposed (n = 12). One slice per area per animal. *** p < 0.001.

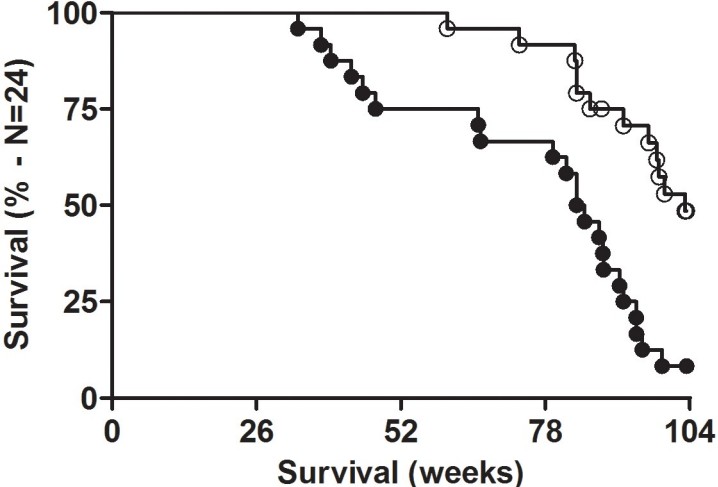

**Fig 2. Survival proportion of rats after repeated daily exposure to SRS source.** Empty circles = sham; black circles = exposed. Survival curves were significantly different (p < 0.001).

group (n = 24) (Fig 2). The median lifespan was 590 days for the exposed group, compared to 722 days for the sham group—p < 0.0001. One sham rat was used as sentinel for sanitary control, eleven sham animals survived at the end of the experiment, whereas only two animals survived in the exposed group. The hazard ratio was 4.1 [CI = 2.0–8.6].

## Lethal tumours and anatomopathology

For tumours identified as the cause of death, eleven of the exposed animals showed one or two large sub-cutaneous tumours of different types (five were malignant—seven appeared before 20 months) (Table 3 and one illustration Fig 3), compared to only two such tumours in the sham group (both malignant, first one at 22 months).

One of the exposed animals with an external tumour also had a pituitary tumour, and at death, six other exposed animals had abdominal masses and one had a pituitary tumour. Tumour types and lifespan are detailed in Table 3 and histological lesions are further listed in S5 Table.

## Discussion

Although they are hugely far above environmental levels, the question has been raised whether intense and very short pulses (nanosecond range) could have health effects. Old studies considered typical radar modulation of 1/1000th (1 µs every millisecond, i.e. repetition rate of 1

**Table 3. Lifespan to death or sacrifice and anatomopathological diagnostic of lesions.**

| Rat # | Age at death (weeks) | Exposed group | Sham group |
|---|---|---|---|
| 4 | 33 | *Internal mass, adenopathies*[a] . . . | |
| 10 | 38 | **Fibrosarcoma**[b] | |
| 21 | 39 | *Internal mass, adenopathies, spleen* | |
| 8 | 43 | **Fibrosarcoma** | |
| 20 | 45 | **Fibroma**, ulcerated | |

*(Continued)*

**Table 3.** (Continued)

| Rat # | Age at death (weeks) | Exposed group | Sham group |
|---|---|---|---|
| 5 | 47 | Posterior limbs paralysed | |
| 36 | 60 | | /c |
| 3 | 66 | *Internal mass, spleen, pancreas* | |
| 6 | 66 | **Subcutaneous adenocarcinoma** | |
| 27 | 73 | | Large cystic kidneys |
| 23 | 79 | **Fibroma**, ulcerated | |
| 17 | 82 | / | |
| 37 | 83 | | / |
| 25 | 84 | | Posterior limbs paralysed |
| 46 | 84 | | / |
| 2 | 84 | *Pituitary tumour* | |
| 11 | 84 | Spongy/granulous kidneys | |
| 19 | 85 | *Mesenteric mass*c, ileum (lysed organs) | |
| 45 | 86 | | *Large preputial glands* |
| 34 | 88 | | Sentinel animal |
| 24 | 88 | Polycystic kidneys | |
| 7 | 88 | **Osteosarcoma** | |
| 16 | 88 | **Fibro-epithelial polypus**, cystic and spongy kidneys | |
| 14 | 91 | *Jejunal mass* | |
| 9 | 92 | Mass: *adrenal/kidney/spleen* (internal bleeding) | |
| 48 | 92 | | *Large preputial glands* |
| 13 | 94 | **Zymbal's gland adenoma**, ulcerated ear area, *pituitary tumour*, cystic and spongy kidneys | |
| 1 | 94 | **Fibro-adenoma**, cystic and spongy kidneys, *tracheo-bronchial ganglions* large and inflammatory | |
| 15 | 95 | / | |
| 29 | 97 | | Large cystic and spongy kidneys, *white lung masses* |
| 35 | 98 | | Cystic and spongy kidneys, duodenum dark and spongy content |
| 26 | 98 | | **Subcutaneous schwannoma**, *large spleen, large left preputial gland* |
| 40 | 99 | | / |
| 12 | 99 | **Fibrosarcoma** Dark abdominal cavity, testes soft small and dark. Soft brain, small spleen, *large left adrenal gland*, external part of lungs grey/brown | |
| 18 | 103 | / | |
| 22 | 103 | **Fibro-adenoma** | |
| 39 | 104 | | **Fibrosarcoma** |
| 28, 30–33 41–44, 47 | 103–104 | | Sacrificed, no tumour |

Left column: exposed group (# = 1–24); right column: sham group (# = 25–48).

[a]*italic*: internal masses found at death or at 24 months;

[b]**bold**: large external masses leading to ethical sacrifice;

[c]/: no macroscopic abnormality. Only two exposed rats survived at the end of the experiment at 103 weeks: #18, 22. Eleven rats of the sham group survived to the end of the experiment: #28, 30–33, 38, 39, 41–44 and 47 were sacrificed at 103 or 104 weeks, 10 of them without any tumour.

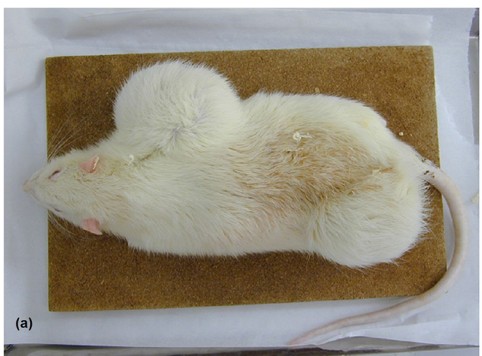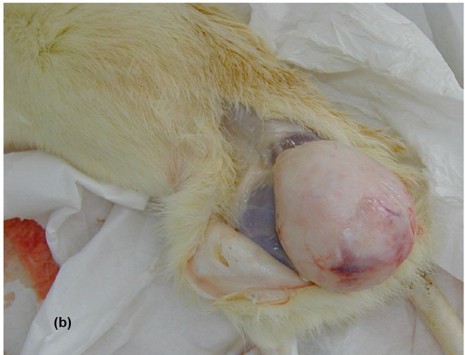

**Fig 3. (a) Picture of one exposed rat with two fibromas (left)–(b) macroscopic view of the femoral tumour (right).** The two fibromas were in the axillary and the femoral area, the macroscopic view was taken at autopsy (death at 18.5 months).

kHz). Below the thermal level of 4 W kg$^{-1}$, no specific effect of modulation had been proven, so this modulation factor of 1/1000$^{th}$ has been considered as safe in the public health standards [20]. Recent studies on high intensity nanosecond pulsed microwaves have been performed on cells and their electrophysiological properties, or on membrane permeability, but none on animals with repeated exposures [16].

## Behavioural tests

Although some stress could be induced by the microwave auditory effect or by possible corona effect following field enhancement near protrusions on animals' bodies (e.g. toes, nose), no effect was observed on learning abilities or anxiety level. However, a positive effect was found in the rotarod test, which mainly assesses sensori-motor activity. This could be due to a slight heating at the SAR of 4.7 W kg$^{-1}$ of the SRS source. Such a heating effect has been hypothesized by Preece who observed an increased reactivity (shorter reaction time) in human volunteers exposed to mobile phones with a local SAR of 1.7 W kg$^{-1}$ [21]. Avoidance of the SRS beam was significant in exposed animals subjected to a thermal SAR of 22 W kg$^{-1}$, which is high above the thermal threshold of 4 W kg$^{-1}$ identified by ICNIRP [20]. This expected result confirms the relevance of this threshold.

## Brain inflammation

The large GFAP level increase in different brain structures after SRS exposure reflects the higher average SAR of 4.7 W kg$^{-1}$ allowed by a continuous emission of SRS pulses instead of spaced 10 s pulse trains with average SAR of 0.34 W kg$^{-1}$ in the SRX. These results are consistent with those previously found with a much lighter modulation of microwaves 1/8$^{th}$ of the time, such as the one produced by GSM mobile phones [22]. In comparison, the modulation of HPM is 1 ns at 100 pps, i.e. a ratio of 10$^{-7}$. Alterations in the glial cell marker GFAP could represent a marker of a long-term risk in rats, but this has yet not been shown. Mainly known as a marker of traumatic injury, GFAP has been considered by previous studies as non-specific, therefore compromising its prognostic power [23].

## Lifespan, lethal tumours and anatomopathology

More importantly, an increased and early rate of sarcomas and fibrosarcomas, as well as higher associated mortality, were observed in animals exposed to repetitive sessions at an average SAR

of 0.8 W kg$^{-1}$, five times below the thermal threshold (Table 1). The spontaneous rate of fibro-sarcomas found in old Sprague Dawley rats at termination of two-year studies is usually around 1 to 3%, and tumours are rarely reported as cause of life shortening [24]. Several studies have tested the impact of HPM, but chronic exposure was only performed with continuous waves, radar-type microwave pulses of the order of microseconds [11,25,26], or mobile phone-type exposures [27]. Effects on cancer and lifespan were reported with SARs close to or above the thermal level [25,28,29]. Only Chou et al. reported an increase in primary malignancies, with-out life shortening, with pulsed waves at low SAR levels (0.15–0.4 W kg$^{-1}$) and exposures lasting 21.5 h/day for 25 months [26]. Recently, a NIEHS study of the National Toxicological Program reported an increased incidence of heart schwannoma and glioma in whole-body exposed rats to phone-type microwaves at much higher SARs than those used in humans [27]. Although experiments with newer extremely short pulses (a few ns long) have been performed, HPM had only been used in acute experiments, and most studies reporting an effect looked at physiological reactions, without addressing genotoxicity or carcinogenicity endpoints. This work there-fore corresponds to the first report with in vivo exposure to extremely short duration peak pulses, with a high repetition rate, and with a design of repeated exposure for eight weeks.

Tinkey showed that very high doses of X-rays (> 46 Gy) were needed to induce sarcomas in Sprague-Dawley rats [30]. To our knowledge, no study showed increased tumour rates in rats exposed to lower doses of X-rays; then the low 0.8 Gy residual X-ray level of this study would probably not explain the observed early tumour increase. This hypothesis would need to be tested with positive controls exposed at 0.8 Gy expanded over the same 2-months period. Therefore, this study shows that the observed tumours and decreased lifespans were due to repeated exposures at a SAR below the known health threshold of 4 W kg$^{-1}$, given that the peak SAR was of the order of 3.3 MW kg$^{-1}$ (E-field above 0.5 MV m$^{-1}$).

Conversely, some studies would support a protective effect of HPM against cancer. Devyat-kov found a decrease in tumour cell proliferation in vitro and an increase in survival time of rats implanted with a liver carcinosarcoma and exposed to 10 ns pulses at 9 GHz, which para-doxically was beneficial [10]. The peak power was 100 MW, but the electric field or SAR was not specified.

More recently, although using UWB pulses which have a different frequency spectrum, after 16–1000 ns pulses with a frequency of 0.6–1.0 GHz, a duration of 4–25 nanosecond, an amplitude of 0.1–36 kV cm$^{-1}$, and a pulse repetition rate of 13 pulses per second (pps), Zhar-kova also found an inhibition of mitochondrial activity which has been interpreted rather as an anti-tumoral activity [31]. However, other studies with UWB bring some mechanistic explanation that could support a cancerogenic effect. Dorsey found an increase in mitogenic activity of mouse hepatocytes [13]. Natarajan published genotoxic effects [32] and Shckorbatov showed some changes in chromatin [33], which studies bring arguments rather in favour of a carcinogenic effect. Whether mechanisms are common between HPM and UWB exposures will have to be further explored.

Observed tumours were mostly subcutaneous, but were also ubiquitous, which is not indic-ative of a specific mechanism or sensitivity of a given tissue or organ. This means that inflam-matory processes or genotoxic effects should be investigated in the different target tissues where tumours appeared: connective tissue, muscle, fat, vessels, pituitary gland, lymph nodes, etc. To check if this is a general phenomenon or a strain/specie specific effect, this experiment should be repeated with other rat strains and different animal species (e.g. mice, and/or rab-bits) which are usual models for human toxicology.

The actual corner stones of guidelines for RF exposures consider behavioural effects as the most sensitive biological endpoint that had yet been observed as a deleterious effect on health. Up to date, this decreased behavioural performance is today attributed to the temperature

elevation produced in rodents or primates, consecutive to the above-mentioned dielectric absorption, only linked to the average absorbed power (rms SAR). This study shows that a repeated and prolonged exposure to extremely high intensity microwave pulses, around one million volts per meter (1 MV m$^{-1}$), comparable to those that have in part been used in the Gulf War, produce a clear increased incidence of cancer in exposed animals. Furthermore, it tells that even an aggressive damage such as cancer can occur without so much decreased cognitive performance, even at a level below the known thermal threshold of whole-body SAR (4 W kg$^{-1}$). Then the peak SAR should be re-considered in the definition of guidelines. However, and luckily, humans are not exposed for such durations at close distance from HPM emitters.

The original hypothesis was: is there an effect of high-power microwaves? In which conditions? If yes, does it obey to a classical thermal mechanism or a mechanism other than thermal? This study showed: i) few behavioural effects from either acute or repeated exposure; ii) an inflammatory effect of acute exposure to HPM; and iii) a surprising increase of lethal cutaneous or subcutaneous tumour incidence of sarcoma or fibrosarcoma type, in the repetitively exposed group (46% versus 8% in the sham-control group). This increased cancer incidence was associated with decreased lifespan in rats exposed to HPM with an average SAR level below the thermal threshold of 4 W kg$^{-1}$. Furthermore, this effect was not associated with clear effects on behaviour, as could have been expected from previous knowledge. The underlying mechanisms are likely to be different from thermal effects and need to be further explored. Also, the thresholds or dose-responses in SAR level, duration and number of exposure sessions need to be defined before establishing limit values for human exposure.

## Supporting information

**S1 Fig. Avoidance test during SRS exposure: % time spent in the shielding box in habituation and exposure periods.**
(TIF)

**S1 Table. Rotarod test data after SRS exposure: Time spent on the rods at training and at test.**
(PDF)

**S2 Table. Avoidance test during SRS exposure: Time and % time spent in the shielding box in habituation and exposure periods.**
(PDF)

**S3 Table. GFAP immunohistochemical labelling after SRX exposure—Raw values.** Magnification x10.
(PDF)

**S4 Table. GFAP immunohistochemical labelling after SRS exposure—Raw values.** Magnification x10.
(PDF)

**S5 Table. Synthesis of histological lesions in sham (Sh) and exposed (Ex) groups after repeated SRS exposure.**
(PDF)

## Acknowledgments

We thank A. Braun for reviewing and editing the paper, and Université de Picardie Jules Verne for financial participation to the publication costs. As the following contributors could

not be contacted for formal approval of the paper, they are acknowledged for their active participation to this work: E. Bourrel: technician, project administrator and investigator of the second half of experiments; L. Caplier: investigator for tissue preparation and anatomopathological analyses; I. Guimiot: supervising and performing immunohistochemical analyses for GFAP, analyses of behavioral studies; O. Dupont: animal technician, housing and care of animals.

## Author Contributions

**Conceptualization:** René de Seze.

**Data curation:** Carole Poutriquet.

**Formal analysis:** Carole Poutriquet.

**Funding acquisition:** René de Seze, Caroline Fonta.

**Investigation:** Carole Poutriquet, Christelle Gamez, Emmanuelle Maillot-Maréchal, Franck Robidel, Anthony Lecomte.

**Methodology:** René de Seze, Carole Poutriquet, Christelle Gamez, Emmanuelle Maillot-Maréchal, Franck Robidel, Anthony Lecomte.

**Project administration:** René de Seze, Caroline Fonta.

**Resources:** Carole Poutriquet, Christelle Gamez, Emmanuelle Maillot-Maréchal, Franck Robidel, Anthony Lecomte.

**Supervision:** René de Seze, Caroline Fonta.

**Validation:** René de Seze, Caroline Fonta.

**Visualization:** René de Seze.

**Writing – original draft:** René de Seze.

**Writing – review & editing:** René de Seze, Carole Poutriquet, Caroline Fonta.

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
