## [Decision Letter · Decision Letter 0]

4 Feb 2020

PONE-D-19-32924

Repeated exposure to nanosecond high power pulsed microwaves increases cancer incidence in rat

PLOS ONE

Dear Dr De Seze,

Thank you for submitting your manuscript to PLOS ONE. After careful consideration, we feel that it has merit but does not fully meet PLOS ONE’s publication criteria as it currently stands. Therefore, we invite you to submit a revised version of the manuscript that addresses the points raised during the review process.

Specifically:

some precisions needs to be given in the material sectionsome modifications are needed in the textsome references could be added to reinforce the message

We would appreciate receiving your revised manuscript by Mar 20 2020 11:59PM. To enhance the reproducibility of your results, we recommend that if applicable you deposit your laboratory protocols in protocols.io, where a protocol can be assigned its own identifier (DOI) such that it can be cited independently in the future. For instructions see: http://journals.plos.org/plosone/s/submission-guidelines#loc-laboratory-protocols

We look forward to receiving your revised manuscript.

Kind regards,

Christophe Egles, Ph.D.

Academic Editor

PLOS ONE

Journal Requirements:

1. At this time, we request that you  please report additional details in your Methods section regarding animal care, as per our editorial guidelines:

(1) Please state whether the provided ethics committee contains animal welfare experts or whether an animal ethics or IACUC committee reviewed and approved the study. Please provide the full name of the committee that reviewed and approved the study  

(2) Please state the number of mice used in the study  

(3) Please provide details of animal welfare (e.g., shelter, food, water, environmental enrichment)

(4) Please describe any steps taken to minimize animal suffering and distress, such as by administering anaesthesia  or analgesia

Thank you for your attention to these requests.

2. To comply with PLOS ONE submission guidelines, in your Methods section, please provide additional information regarding your statistical analyses. For more information on PLOS ONE's expectations for statistical reporting, please see https://journals.plos.org/plosone/s/submission-guidelines.#loc-statistical-reporting

Reviewers' comments:

Reviewer's Responses to Questions

**Comments to the Author**

1. Is the manuscript technically sound, and do the data support the conclusions?

Reviewer #1: Yes

Reviewer #2: Yes

2. Has the statistical analysis been performed appropriately and rigorously? 

Reviewer #1: Yes

Reviewer #2: Yes

3. Have the authors made all data underlying the findings in their manuscript fully available?

Reviewer #1: Yes

Reviewer #2: Yes

4. Is the manuscript presented in an intelligible fashion and written in standard English?

Reviewer #1: Yes

Reviewer #2: Yes

5. Review Comments to the Author

Reviewer #1: Worthy but small study by an experienced group that explores new grounds in RF bioeffects. Several comments

1. Please describe the exposure facility better, Were any conductive objects (e.g. metal tubes from water bottles) in the field? Describe the waveforms of the pulses (I assume they were gated sinewave with a spectral range that is small compared to the carrier frequency - is that correct?) Were the animals "chipped"? (a potential source of enhanced local exposure within the body)

The exposures are extraordinarily large (MV/m) and not far from the breakdown threshold for air. This makes me wonder if corona was involved. There will be field enhancement near protrusions on animals' bodies (e.g. toes, nose), possible sites for corona which would be a source of stress to the animals. Would the authors comment on this as a possible mechanism for the observed effects.

2. The animals clearly found the stimulus to be obnoxious (avoided exposure when given the choice). I note that the fluence (of the order of tens of J/m^2) is far higher than the threshold for the microwave auditory effect (tens of millijoules/m^2). So the animals were undoubtedly perceiving strong audible sensations from the exposure. Is there any evidence for this as a source of stress?

less major:

Writing is imprecise.

1. It is not correct to compare the present study (pulse modulated microwaves) with those using ultrawideband pulses (e.g. Ref. 16; p. 17). The latter have a different frequency spectrum including in some cases a DC component, which will produce quite different effects. Please avoid this confusion.

2. Authors repeatedly refer to "SAR". Please make it clear every time whether they refer to whole body SAR or partial body SAR in an identified region of the body, or spatially peak SAR at any place in the body. Preece (ref 20) was referring to partial body SAR from a phone in localized regions of the head averaged over time, no whole body SAR as referred to in this study.

3. p. 19 "comparable to those that have in part been used in the Gulf War" - were humans exposed to RF pulses comparable in strength to those in this study? For how long and under what circumstances?

Reviewer #2: The authors present a very interesting and important paper, which has to be published after some amendments. These includs some modification as well as more information within the Materials and Methods.

• Legends to the figures are missing

• Were the experiments performed in a blinded manner?

• Regarding the GFAP-staining, it is not clear how this protocol was applied. Here clear information need to be included (which kind of antibody was used, how it was applied, concentration, incubation time etc.).

• In the discussion (line 233) the authors cite a single one author, namely Tinkey (29) as the one proving that >46 Gy X-ray is needed to induce sarcomas in rats. They write that this the prove that 0.8 Gy cannot be the reason for the increased tumor rate. A single study cannot be used as a “prove”, so please change this sentence accordingly. It could have been helping using positive controls at 0.8 Gy level.

• In Table 2: please change in second row “10mn” to “10 min”; and in the 4th row “Nb” to “N”

• There are also some more typographic errors and some linguistic problems that need to be improved

6. PLOS authors have the option to publish the peer review history of their article (what does this mean?). If published, this will include your full peer review and any attached files.

Reviewer #1: Yes: Kenneth Foster

Reviewer #2: Yes: Myrtill Simko

---

## [Author Response · Author response to Decision Letter 0]

27 Feb 2020

Response to reviewers

Editor:

1. At this time, we request that you please report additional details in your Methods section regarding animal care, as per our editorial guidelines:

(1) Please state whether the provided ethics committee contains animal welfare experts or whether an animal ethics or IACUC committee reviewed and approved the study. Please provide the full name of the committee that reviewed and approved the study 

At the time of the study (2005-2007), the ethics committees were not formalized in France. The ethics file has been submitted to the INERIS ethics committee, based on experimented animal technicians and on good laboratory practices labelled at INERIS. The study was named “MFP study”, but not numbered. 

(2) Please state the number of rats used in the study 

Altogether, 294 rats were used in this published part of the study. Numbers have been given page 7 lines 76, and page 8 lines 91-92 for exposed (144), sham (144) and test (6) animals. 

(3) Please provide details of animal welfare (e.g., shelter, food, water, environmental enrichment)

Rats were accustomed to the animal facility during 5 days before handling. They were kept under specific pathogen free (SPF) controlled environmental conditions (ambient temperature 22±1 ◦C, 12-h light/12-h dark cycle) and received food and tap water ad libitum, except during exposure. They were housed in groups of 2 animals per enriched cage for social comfort. Enrichment consisted in adding corn chips for animals to nest and play. The day before exposure, rats were handled and accustomed to the experimenter. This has been added page 7 lines 70-75. 

(4) Please describe any steps taken to minimize animal suffering and distress, such as by administering anaesthesia or analgesia

No experimental procedure was painful and needed analgesia. Upon sacrifice before biochemical analyses, to avoid pain and distress, animals were anaesthetized with an overdose of pentobarbital (i.p.; 50 mg kg-1) until brain-dead stage. This has been added on page 7, lines 83-84. 

2. To comply with PLOS ONE submission guidelines, in your Methods section, please provide additional information regarding your statistical analyses. For more information on PLOS ONE's expectations for statistical reporting, please see https://journals.plos.org/plosone/s/submission-guidelines.#loc-statistical-reporting

The used softwares have been defined (page 13 line 168), and results have been further defined (mean) and completed with SD where useful (behavioural tests page 14, lines 181-183 and 188-190).

 

Reviewer #1: Worthy but small study by an experienced group that explores new grounds in RF bioeffects. Several comments

1. Please describe the exposure facility better, 

The rats were transferred from the animal facility through an airlock to the shielded exposure room 50m from there in plastic boxes with filtering covers. A thin metallic grid supported the filtering cover. The cages were set on a plastic support for acute exposure just in front of and perpendicular to the emitting horn of the HPM generator. Then exposure came to the animal from the side. Details have been added page 8 lines 88-90.

Were any conductive objects (e.g. metal tubes from water bottles) in the field? 

The food grid and the water bottles were removed from the cages for the exposures. 

Describe the waveforms of the pulses (I assume they were gated sinewave with a spectral range that is small compared to the carrier frequency - is that correct?) 

At 10 GHz, the SRX source emits about 10 oscillations of 1 ns. Spectrum width is about 10%, i.e. 1 GHz (inverse of the duration, obtained by Fourier Transform). What about the rise-time and fall-time? Rise-time and fall-time are about 0.2 ns.

At 3.7 GHz, the SRS source emits about 9 oscillations of 2.5 ns. Spectrum width is about 400 MHz (inverse of the duration, obtained by Fourier Transform). What about the rise-time and fall-time? Same as for SRX? Rise-time and fall-time are about 0.2 ns.

Details have been added page 5 lines 63-64. 

Were the animals "chipped"? (a potential source of enhanced local exposure within the body)

The animals were not chipped but labelled with a marker pen for the duration of the exposure period, and ears were tattooed for the lifetime follow-up. Then there was no metallic component close to the rat’s body. This has been added page 7 lines 69-70. 

The exposures are extraordinarily large (MV/m) and not far from the breakdown threshold for air. This makes me wonder if corona was involved. There will be field enhancement near protrusions on animals' bodies (e.g. toes, nose), possible sites for corona which would be a source of stress to the animals. Would the authors comment on this as a possible mechanism for the observed effects.

We thank the reviewer for this interesting comment. Indeed, at this field level, a field enhancement near protrusions on animals' bodies (e.g. toes, nose) is possible, and corona effect could be a source of stress to the animals. However, the cognitive tests measuring anxiety (T-Maze, open-field) did not show any change of anxiety level in exposed rats compared to sham-controls.

This concern has been addressed together with the next one: one sentence and precision have been added page 19 lines 231-234

2. The animals clearly found the stimulus to be obnoxious (avoided exposure when given the choice). I note that the fluence (of the order of tens of J/m^2) is far higher than the threshold for the microwave auditory effect (tens of millijoules/m^2). So, the animals were undoubtedly perceiving strong audible sensations from the exposure. Is there any evidence for this as a source of stress?

The avoidance test was the only one with the maximum thermal exposure (22 W/kg). We agree with the reviewer that the exposure level in our experiments (20 J/m² with the SRX source, 2 J/m² with the SRS source) is higher than the threshold for the auditory effect (100-400 mJ/m²). However, the auditory effect is described to be perceivable between 200 MHz and 6.5 GHz (ICNIRP, 1998), then it should not be detected with the SRX source, and it is potentially strongly perceived with the SRS source, 5-fold above the threshold level. Although this is a possible source of stress, there again cognitive tests did clearly not show increased anxiety in exposed rats compared to sham-controls.

To address these two last concerns, one sentence and precision have been added page 17 lines 231-234

less major:

Writing is imprecise.

1. It is not correct to compare the present study (pulse modulated microwaves) with those using ultrawideband pulses (e.g. Ref. 16; p. 17). The latter have a different frequency spectrum including in some cases a DC component, which will produce quite different effects. Please avoid this confusion.

When UWB have been used, clarifications have been made in the manuscript: 

On page 3, the paper by Dorsey has been individualized (lines 37-38), and it has been specified that reviews by Seaman and Shunck focused on UWB (line 40). 

On page 22, it has been specified when studies have been performed with UWB exposures: Dorsey, Zharkova, Natarajan and Shckorbatov (lines 282-290). 

2. Authors repeatedly refer to "SAR". Please make it clear every time whether they refer to whole body SAR or partial body SAR in an identified region of the body, or spatially peak SAR at any place in the body. Preece (ref 20) was referring to partial body SAR from a phone in localized regions of the head averaged over time, no whole-body SAR as referred to in this study.

SAR has been defined in this study as whole-body SAR (page 3 line 31) and peak SAR as temporal peak-SAR (page 3 line 33, page 13 line 160). Where different, this has been pointed out (Preece’s study: with a local SAR of …; page 18 line 237)

3. p. 19 "comparable to those that have in part been used in the Gulf War" - were humans exposed to RF pulses comparable in strength to those in this study? For how long and under what circumstances?

In the context referring to Gulf War the pulses were not directed to humans at such a close distance. The emissions were shorter, as it does not take long to inhibit electronics. 

The addressed question is: do HPM exposures induce cancer? At which intensity and which duration? 

After the results of this study, a dose-response relationship must be established, and once threshold for a risk will have been defined, safety margins will have to be chosen to protect exposed humans (operators or in targeted vehicles). 

To avoid a too quick extrapolation to human, it has been added on pages 22-23:

Lines 302-303: “This study shows that a repeated and prolonged exposure to extremely high intensity microwave pulses”

Lines 308-309: “However and luckily, humans are not exposed for such durations at close distance from HPM emitters.”

Lines 320-321: duration and number of exposure sessions need to be defined “before to allow establishing limit values for human exposure.”

Reviewer #2: The authors present a very interesting and important paper, which has to be published after some amendments. These include some modification as well as more information within the Materials and Methods.

• Legends to the figures are missing

Following PLOS One instructions to authors, we placed figure captions “in the manuscript text in read order, immediately following the paragraph where the figure is first cited. Do not include captions as part of the figure files or submit them in a separate document.”

• Were the experiments performed in a blinded manner?

For behavioural tests and biochemical analyses, the experiments were blinded (added line 112 page 10). 

For the two-year follow-up, the exposed and the sham group had been identified. 

• Regarding the GFAP-staining, it is not clear how this protocol was applied. Here clear information needs to be included (which kind of antibody was used, how it was applied, concentration, incubation time etc.).

Thanks for asking these precisions. We did not want to be too long in methods description, but here is our detailed protocol:

• Brain preparation

Animals were anaesthetized with pentobarbital (i.p.; 50 mg kg-1) until brain-dead stage 2 or 7 days after exposure. An intra-cardiac perfusion was then performed with a 0.9% NaCl solution, followed by a 4% paraformaldehyde solution in 0.1 M phosphate buffer (pH 7.6) at a flow rate of 30 ml min-1. The brain was dissected and immerged in a 30% sucrose/4% paraformaldehyde solution for 48 h under agitation at 4 ◦C to cryo-protect the cerebral tissue. Using a cryostat microtome, 40 µm-thick sagittal brain slices were obtained and processed as free-floating sections for GFAP immunodetection [17]. 

• Immunohistochemistry 

Brain slices were washed in phosphate buffer saline (PBS; pH 7.6) 3×20 min and incubated for 30 min at room temperature in 1% H2O2 to block endogenous peroxidases. Background noise was saturated by incubation for one hour in a solution of goat serum (5%), bovine serum albumin (2%) and triton (0.3% Triton). Slices were then incubated overnight at 4 ◦C with the primary polyclonal antibody anti-GFAP (rabbit antibody, 1/8000 in PBS. On the second day, the slices were washed with PBS-triton (3×20 min) and incubated for 1 h with the secondary antibody (biotinylated goat anti-rabbit, ABC kit 1/500 in PBS; pH 7.6). Sections were washed in PBS (3x20 min; pH 7.6). Revelation of immunostaining was revealed with a ABC/VIP kit (Vectastain, Vector). Slices were mounted on slides, dehydrated then mounted with DPX before image analysis. Surfaces of labelled areas have been quantified using an optical microscope (Zeiss) coupled with a Colour Camera 3 Charge Coupled Device (CCD) (Sony) and the Visilog 6.2 (NOESIS society, Les Ulis, France) analyser system.

The protocol has been described page 11-12 lines 123-142.

• In the discussion (line 233) the authors cite a single one author, namely Tinkey (29) as the one proving that >46 Gy X-ray is needed to induce sarcomas in rats. They write that this proves that 0.8 Gy cannot be the reason for the increased tumour rate. A single study cannot be used as a “prove”, so please change this sentence accordingly. It could have been helping using positive controls at 0.8 Gy level.

It is right that a single study cannot be a proof and that it could have been helping using positive controls at 0.8 Gy level. However, we carefully looked at the literature and did not find that lower levels of X-rays could cause an increase of early tumour rates during the lifetime of the animals. Usually, studies with X-rays show increase of tumours only at the anatomopathological examination of the animals post-mortem. However, to satisfy this comment, text has been changed accordingly (page 21, lines 271-274): “To our knowledge, no study showed increased tumour rates in rats exposed to lower doses of X-rays; then the low 0.8 Gy residual X-ray level of this study would probably not explain the observed early tumour increase. This hypothesis would need to be tested with positive controls exposed at 0.8 Gy expanded over the same 2-months period.”

• In Table 2: please change in second row “10mn” to “10 min”; and in the 4th row “Nb” to “N”

Done

• There are also some more typographic errors and some linguistic problems that need to be improved

The whole paper has carefully been reread and corrections have been made where felt to be needed.

---

## [Editor Report · Decision Letter 1]

18 Mar 2020

Repeated exposure to nanosecond high power pulsed microwaves increases cancer incidence in rat

PONE-D-19-32924R1

Dear Dr. De Seze,

We are pleased to inform you that your manuscript has been judged scientifically suitable for publication and will be formally accepted for publication once it complies with all outstanding technical requirements.

With kind regards,

Christophe Egles, Ph.D.

Academic Editor

PLOS ONE
---

## [Editor Report · Acceptance letter]

26 Mar 2020

PONE-D-19-32924R1 

Repeated exposure to nanosecond high power pulsed microwaves increases cancer incidence in rat 

Dear Dr. De Seze:

I am pleased to inform you that your manuscript has been deemed suitable for publication in PLOS ONE. Congratulations! Your manuscript is now with our production department. 

With kind regards,

on behalf of

Professor Christophe Egles 

Academic Editor

PLOS ONE